# Tenascin-C in Tissue Repair after Myocardial Infarction in Humans

**DOI:** 10.3390/ijms241210184

**Published:** 2023-06-15

**Authors:** Kenta Matsui, Sota Torii, Shigeru Hara, Kazuaki Maruyama, Tomio Arai, Kyoko Imanaka-Yoshida

**Affiliations:** 1Department of Pathology and Matrix Biology, Graduate School of Medicine, Mie University, 2-174 Edobashi, Tsu 514-8507, Japan; k-matsui@med.mie-u.ac.jp (K.M.);; 2Department of Pathology, Tokyo Metropolitan Geriatric Hospital and Institute of Gerontology, 3-52 Sakaecho, Itabashi-ku, Tokyo 173-0015, Japan

**Keywords:** inflammation, myocardial infarction, ventricular remodeling, lymphatic system, angiogenesis

## Abstract

Adverse ventricular remodeling after myocardial infarction (MI) is progressive ventricular dilatation associated with heart failure for weeks or months and is currently regarded as the most critical sequela of MI. It is explained by inadequate tissue repair due to dysregulated inflammation during the acute stage; however, its pathophysiology remains unclear. Tenascin-C (TNC), an original member of the matricellular protein family, is highly up-regulated in the acute stage after MI, and a high peak in its serum level predicts an increased risk of adverse ventricular remodeling in the chronic stage. Experimental TNC-deficient or -overexpressing mouse models have suggested the diverse functions of TNC, particularly its pro-inflammatory effects on macrophages. The present study investigated the roles of TNC during human myocardial repair. We initially categorized the healing process into four phases: inflammatory, granulation, fibrogenic, and scar phases. We then immunohistochemically examined human autopsy samples at the different stages after MI and performed detailed mapping of TNC in human myocardial repair with a focus on lymphangiogenesis, the role of which has recently been attracting increasing attention as a mechanism to resolve inflammation. The direct effects of TNC on human lymphatic endothelial cells were also assessed by RNA sequencing. The results obtained support the potential roles of TNC in the regulation of macrophages, sprouting angiogenesis, the recruitment of myofibroblasts, and the early formation of collagen fibrils during the inflammatory phase to the early granulation phase of human MI. Lymphangiogenesis was observed after the expression of TNC was down-regulated. In vitro results revealed that TNC modestly down-regulated genes related to nuclear division, cell division, and cell migration in lymphatic endothelial cells, suggesting its inhibitory effects on lymphatic endothelial cells. The present results indicate that TNC induces prolonged over-inflammation by suppressing lymphangiogenesis, which may be one of the mechanisms underlying adverse post-infarct remodeling.

## 1. Introduction

Acute myocardial infarction (MI) results from severe ischemia due to coronary arterial occlusion/obstruction, most typically by atherothrombosis, and is the leading cause of morbidity and mortality worldwide [1,2]. Timely reperfusion by percutaneous coronary intervention (PCI) or thrombolytic therapy very effectively reduces the infarct size and significantly improves the short-term prognosis of patients. After acute-phase treatment, the infarct segment of the ventricular wall is gradually stretched and thinned, resulting in an increase in the chamber size within 30 days during the healing stage. However, in several cases, the chamber size progressively increases for weeks or months after MI, resulting in systolic dysfunction and aggressive progression to heart failure, which is clinically referred to as post-infarct remodeling. This long-term ventricular remodeling is currently regarded as the most critical sequela of MI [1,2]. Adverse post-infarct remodeling is explained by inadequate tissue repair due to dysregulated inflammation during the acute phase; however, its pathophysiology remains unclear. Extensive clinical attempts have been made to identify diagnostic indicators and therapeutic target molecules, and tenascin-C (TNC) has been proposed as a plausible candidate [3,4].

Tenascins are a family of multimeric extracellular matrix (ECM) molecules with a domain structure that is characterized by an N-terminal globular domain, a series of tenascin-type epidermal growth factor (EGF)-like repeats, a series of fibronectin type III domains, and a C-terminal fibrinogen-related domain [5,6]. Four tenascin members have been identified to date: TNC, tenascin-R, tenascin-X (known as tenascin-Y in birds), and tenascin-W. TNC was the first tenascin identified and is also an original member of the matricellular protein family together with thrombosondin-1 and secreted protein acidic and rich in cysteine (SPARC; osteonectin) [7,8].

As a typical matricellular protein, TNC is highly up-regulated by changes in the tissue organization structure, such as embryogenesis, inflammation, tissue repair, regeneration, or cancer invasion, but it is expressed at low levels in normal adults [9,10]. One of the features of TNC is its restricted spatio-temporal expression pattern, which is particularly evident in the heart. In the embryogenic heart, TNC is transiently expressed at specific sites in several important steps, such as the differentiation of precardiac mesodermal cells or the development of coronary arteries [11]. When the heart is formed, TNC expression is rapidly down-regulated and is not detected in normal adults. However, it is highly up-regulated in various myocardial disease states, such as MI, myocarditis, and hypertensive fibrosis, and this is closely associated with active inflammation. While TNC molecules are deposited in the interstitium of myocardial lesions, a soluble fraction exists and is released into the blood; therefore, TNC is used as a diagnostic blood biomarker. In patients with MI, TNC blood levels increase within 24 h of the onset, peak on the fifth day, gradually decrease, and then normalize within approximately one month. Higher peak levels of TNC on day five predict an increased risk of ventricular dilatation three months after the onset of MI and a poorer prognosis at 50 months [3]. Therefore, the effects of TNC during the acute stage after MI have an impact on the progression of ventricular remodeling in the chronic stage.

MI in adult mammals heals with scar formation because of the limited regenerative capacity of cardiomyocytes. The healing process consists of a complex series of cellular activities. Ischemic cell injury and necrosis induce intense sterile inflammation and inflammatory cell infiltration, which clear damaged tissue. Granulation tissue then forms in the affected area by neovascularization, the recruitment and proliferation of fibroblasts/myofibroblasts, and matrix synthesis associated with inflammatory cells [12]. With the resolution of inflammation, cellular components and the neovasculature eventually decrease, and the damaged myocardium is replaced by collagen-rich scar tissue. Experimental TNC-deficient or -overexpressing mouse models indicate that TNC synthesized by local interstitial cells affects many different cell types in an autocrine or paracrine manner to regulate diverse and often opposing cellular responses in a context-dependent manner [13,14,15]. Accumulating evidence has shown the proinflammatory effects of TNC on macrophages [13,16,17,18,19,20,21,22]. Although inflammation is essential for myocardial repair, dysregulated over-inflammation may lead to adverse post-infarct remodeling. Recently, the roles of lymphatic vessels in enhancing the clearance of excessive interstitial fluids, proinflammatory mediators, and immune cells [23,24,25,26,27,28] have been attracting increasing attention as one of the mechanisms to resolve inflammation.

The present study investigated the roles of TNC in human myocardial repair, namely, which cell activities are reflected by the elevated level of TNC and may be responsible for chronic adverse remodeling. The detailed mapping of TNC molecules may provide insights into the specific cellular targets for and activities of TNC based on its restricted spatio-temporal localization [29]. Therefore, we herein examined human autopsy samples at different stages after MI with immunostaining and analyzed the localization of TNC during human myocardial repair with a focus on lymphangiogenesis. Furthermore, the direct effects of TNC on lymphatic endothelial cells (LECs) were investigated using RNA sequencing.

## 2. Results

### 2.1. Expression Mapping of TNC during Tissue Repair after MI in Humans

By modifying the staging proposed by Franngogianis [2,12], we categorized histopathological findings of 20 cases (characteristics of each case are shown in Table 1) into four phases based on the appearance of inflammatory cells, angiogenesis, and collagen fibers: the inflammatory phase, granulation phase, fibrogenic phase, and scar phase.

The inflammatory phase was defined by the coagulative necrosis of cardiomyocytes associated with the infiltration of primarily neutrophils. The feature of the granulation phase was angiogenesis in which CD31-positive blood endothelial cells proliferate and form a vascular structure as well as the prominent infiltration of macrophages instead of neutrophils, but few αSMA-positive myofibroblasts. The fibrogenic phase was characterized by numerous αSMA-positive myofibroblasts and the maturation of collagen fibers. The scar phase was defined by the deposition of mature thick collagen fibers with the disappearance of myofibroblasts, blood vessels, and inflammatory cells. However, an autopsy specimen from one patient showed lesions at various time phases of healing. Therefore, we carefully compared the immunolocalization of TNC with the location and appearance of macrophages, vascular endothelial cells, LECs, and formation of collagen fibrils.

*Inflammatory phase* (Figure 1)*:* In the infarcted area, cardiomyocytes had an eosinophilic cytoplasm and no nuclei. The interstitium was wide and edematous. The infiltration of neutrophils, but not macrophages, was noted. Immunostaining for TNC was diffusely observed in the infarcted area and border zone. CD68-positive macrophages and podoplanin-positive lymphatic vessels were not detected in the infarcted area.

*Early granulation phase:* As dead cells and matrix debris were cleared by infiltrating macrophages, endothelial cells sprouted from the preexisting vascular vessels in the intact myocardium and formed vascular channels in the loose provisional matrix in the lesion. Then, αSMA-positive myofibroblasts appeared at the border zone and migrated into nascent granulation tissue. Distinct immunostaining for TNC was observed in the interstitium at the border zone between the affected area and residual myocardium, at which angiogenic sprouting and macrophage infiltration were noted (Figure 2A). Neither podoplanin-positive lymphatic vessels nor the formation of collagen fibers was detected (Figure 2B) at this stage. A few podoplanin-positive cells appeared after the formation of vascular vessels and often appeared to be associated with new blood vessels.

*Fibrogenic phase:* Many αSMA-positive myofibroblasts were distributed throughout the infarct lesion, and some were associated with the walls of new blood vessels. Podoplanin-positive lymphatic tubes formed at the sites of CD31-positive vascular vessels (Figure 3A). Although the CD31 and podoplanin staining of tubular structures generally did not overlap, some tubules appeared to be positive for both CD31 and podoplanin. Thin and short collagen fibers became detectable at the sites at which vascular and lymphatic vessels formed (Figure 3B). TNC staining decreased and was localized to the vicinity of the residual myocardium and part of the granulation tissue that replaced the infarct lesion. Lymphatic vessels and relatively mature thick collagen fibers were primarily observed in TNC-negative areas.

*The scar phase* (Figure 4): Significant decreases were observed in the numbers of macrophages, myofibroblasts, and vascular and lymphatic vessels. The infarct area was filled with mature, thick collagen fibers. Collagen fibers extended between cardiomyocytes in the intact area. TNC immunostaining was not observed at this stage.

### 2.2. Sequential Changes in Blood Vessel and Lymphatic Vessel Densities

We performed a semi-quantitative analysis of sequential changes in vascular angiogenesis and lymphangiogenesis. Many blood capillaries were observed between cardiomyocytes in the intact myocardium, whereas only a few lymphatic vessels were detected (Figure 5A(a–c)). In the inflammatory phase, neither blood nor lymphatic vessels were observed in the infarct area (Figure 5A(d–f)). The density of new blood vessels increased in the granulation phase but significantly decreased in the fibrogenic phases. Lymphatic vessel numbers began to increase in the granulation phase, peaked in the fibrogenic phase, and decreased in the scar phase (Figure 5B).

### 2.3. RNA Sequencing

We examined TNC-mediated transcriptional targets that may be involved in suppressing lymphangiogenesis during the pathogenesis of MI. An RNA-seq analysis was performed on TNC-treated and control LECs. In comparison with control LECs, the transcripts of seven genes were up-regulated, whereas those of four genes were down-regulated in TNC-treated LECs (Figure 6A–C). In addition, a gene ontology enrichment analysis using ShinyGO 0.772 revealed significantly affected categories in genes that were down- or up-regulated in response to the TNC treatment. Down-regulated genes were associated with nuclear division, cell migration, cell motility, and cell cycles (Figure 6D), suggesting that TNC is closely involved in the proliferation and migration of LECs. While the biological significance of these results remains unclear, up-regulated genes were associated with targeting proteins to the cell membrane.

## 3. Discussion

Myocardial tissue repair after MI progresses in a time-dependent manner and is often divided into three phases: the inflammatory phase, proliferative phase, and maturation phase. Generally, in humans, the inflammatory phase lasts up to four days, and the proliferative phase occurs from days 5 to 14 after MI [2,12]. Comparisons of this time course with clinical data on serum TNC levels in MI patients indicated that TNC is up-regulated in the early stage of the proliferative phase and appears to play a role in healing. However, the underlying mechanisms remain unclear because the repair process involves a complex and overlapping series of cellular activities. Furthermore, since reperfusion therapy exerts a number of effects on cellular responses, difficulties are associated with interpreting the biological roles of TNC.

To simplify cellular responses after MI, we herein examined autopsy samples from MI patients who did not receive reperfusion therapy. Furthermore, we subdivided the “proliferative phase” into a granulation phase and a fibrogenic phase to assess the specific cellular activities during tissue repair. The mapping of TNC expression supported its potential role in the regulation of macrophages, sprouting angiogenesis, recruitment of myofibroblasts, and early collagen fibril formation during the inflammatory phase to the early granulation phase of human myocardial repair, which have been suggested using experimental animal models. For example, TNC promoted the migration and differentiation of myofibroblasts, and the recruitment of myofibroblasts was delayed in TNC knockout mice [30]. TNC was also identified as a profibrotic molecule [9,10,31] that may stabilize and integrate ECM fibrils by binding to fibronectin and type I mediated by periostin [32]. Accumulating evidence has shown that TNC augments the pro-inflammatory activity of macrophages via TRL4 [13,18,19,20] or integrin αvβ3 [22]. Macrophages are major inflammatory cells in tissue repair after MI and are broadly classified as M1 or M2 macrophages. The present study showed the prominent deposition of TNC around macrophages in the early granulation phase. Previous human autopsy studies reported that the monocyte-derived macrophages that accumulated during the first five days (inflammatory phase) were mainly proinflammatory M1 macrophages, whereas most macrophages have the anti-inflammatory M2 phenotype when granulation tissue is actively formed between days 5 and 20 after MI [33,34]. M2 macrophages secrete anti-inflammatory, angiogenic, and profibrotic cytokines, such as TGF-β and VEGF, in order to terminate inflammation, and drive angiogenesis and collagen deposition to promote tissue repair, whereas M1 macrophages are also crucial for effective healing through the clearance of necrotic and apoptotic cells. The transition from M1 to M2 macrophages may be a key process for the resolution of inflammation and adequate tissue repair. The present study showed the prominent deposition of TNC around macrophages in the early granulation phase in humans. We previously reported that TNC enhanced M1 polarization but inhibited M2 polarization [13] in mouse models; therefore, TNC may play a role in the M1/M2 transition in the early granulation phase.

Moreover, the present results suggest that TNC supports vascular angiogenesis but suppresses lymphangiogenesis.

The generation of new capillary beds is essential for tissue healing [35]. Neovascularization generally involves angiogenesis and vasculogenesis. Vasculogenesis is the process of de novo blood vessel formation by newly differentiated endothelial cells. Angiogenesis is the process of growing new blood vessels and is initiated by sprouting capillaries from the existing vasculature. It consists of several steps: the separation of pericyte, the migration and proliferation of endothelial cells, remodeling into capillary tubes, and the recruitment of pericytes/smooth muscle cells [36,37,38].

Neovascularization during myocardial repair is initiated by sprouting angiogenesis from existing blood vessels at the border zone of the intact myocardium, which is a major component of granulation tissue. Neovascular tubules are formed and mature by recruiting pericytes/smooth muscle cells but subsequently disappear with the maturation of scar tissue. In human samples in the early stage after MI, the immunolocalization of TNC suggests its involvement in the early activity of angiogenesis in myocardial repair. Several roles for TNC in cardiac angiogenesis have been reported. A heart transplantation model indicated that TNC modulated responses to the angiogenic growth factors of endothelial cells and the homing of bone marrow-derived endothelial progenitor cells [39]. TNC has also been suggested to promote the recruitment of pericytes to developing coronary arteries via PDGF-BB/PDGF-Rβ in the embryo [40,41].

A number of experiments have established that TNC exerts pro-angiogenic effects under a number of conditions [42], such as after cornea cauterization [43], ischemic proliferative retinopathy [44], and wound healing [45]. VEGF-A/VEGFR2 signaling has been shown to induce the migration of tip cells from the preexisting vasculature (reviewed in [46]) in angiogenesis. Since TNC binds to VEGF-A [47,48], it may regulate the migration and proliferation of endothelial cells.

The relationship between TNC and blood vessels has been extensively examined in cancer stroma [49,50,51,52], and its role in the formation of vessel-like structures has been reported in gastric cancer [53] and glioma [54]. Research is gradually revealing aspects of the complex control systems of cancer angiogenesis. TNC is highly expressed in the so-called angiomatrix, a matrisomal signature that characterizes the angiogenic switch to enhance angiogenesis [55,56]. It has been shown to promote angiogenesis via Wnt signaling by down-regulating the Wnt inhibitors Dickkopf-1 (DKK1) [49] and ephrin-B2/EPHB4, and also induces anti-angiogenic signaling via the down-regulation of YAP [56]. Furthermore, TNC promotes tubulogenic activity by inducing FN expression in endothelial cells [57]. Therefore, TNC may be a regulator of angiogenic cellular signaling more than a simple pro-angiogenic molecule. While lymphatic vessels share some morphological similarities with blood vessels, they have unique hierarchical network structures composed of thin-walled open-ended capillaries and collecting vessels that transport lymph via the lymph nodes towards the thoracic ducts and then into the superior vena cava [24,28].

The cardiac lymphatic network spans all layers of the ventricles and drains into a subendocardial plexus, which plays an important role in maintaining the tissue fluid balance, immune surveillance, and immune cell trafficking [28,58]. Cardiovascular lymphatic vessels have a heterogeneous origin. Between 50% and 60% of LECs on the ventral side of the heart originate from the cardiopharyngeal mesoderm (CPM), a well-established source for cranial and myocardial muscle development. CPM-derived lymphatic vessels are formed via a process referred to as lymphvasculogenesis by the direct transdifferentiation of mesodermal cells into LECs. In contrast, the dorsal component of cardiac LECs is derived from the common cardinal vein through lymphangiogenesis. Moreover, it has been postulated that lymphatic endothelial cells may additionally originate from yolk sac-derived hemogenic endothelial cells, albeit to a lesser degree [59,60,61].

The adult lymphatic vasculature is mostly quiescent under physiological conditions; however, in pathological states, such as inflammation, lymphatic vessel remodeling is promoted through lymphangiogenesis and/or changes in lymphatic functions [23,62]. MI has been shown to reactivate lymphangiogenesis as part of the attendant inflammatory response [59,63,64,65,66].

The remodeled cardiac lymphatic network facilitates the clearance of excessive interstitial fluid, debris, proinflammatory mediators, and excess immune cells [23]. It also promotes tissue remodeling and wound healing [26,67,68,69]. The augmentation of lymphangiogenesis has been shown to not only limit myocardial edema and inflammation in the acute stage but also reverse the progression of heart failure in the chronic stage [23,25,70,71,72,73,74,75,76].

In adults, lymphangiogenesis mainly occurs through the sprouting of new lymphatic vessels from the existing lymphatic system [77,78,79,80]. Under physiological and pathological conditions, the growth of lymphatic vessels is associated with angiogenesis [69,81]. However, lymphangiogenesis occurs independently of hemangiogenesis and in the absence of preexisting blood vessels, while blood vessels may grow independently of lymphatic vessels [82]. Furthermore, increases in lymphatic vessels often occur after neovascularization [65], which was confirmed in the present study.

The regulatory mechanisms of lymphangiogenesis appear to overlap in some parts but differ from those of vascular angiogenesis. VEGFs, particularly VEGF-C and VEGF-D, induce the most potent and specific prolymphangiogenic signaling through VEGFR3 [83]. Other inflammation-related signaling pathways, such as tumor necrosis factor-α [84], lymphotoxin-α [85], toll-like receptor signaling [86], NF-κB [87], erythropoietin [88], COX-2 [89], and prostaglandin E2 signaling [90], stimulate lymphangiogenesis [69]. In contrast, limited information is available on the negative regulation of lymphangiogenesis. IFN-γ/JAK/STAT [91], TGFβ [92,93,94], endostatin [95], thrombospondin [96], and semaphorin3E-PlexinD1 signaling [71] suppress lymphangiogenesis.

In the present study, new lymphatic vessels were observed in relatively mature granulation tissue, in which TNC immunostaining was reduced. Furthermore, our RNA-seq analysis revealed that TNC modestly down-regulated genes related to nuclear division, cell division, and cell migration in LECs. Consequently, TNC may exert direct inhibitory effects on LECs, thereby suppressing lymphangiogenesis.

In conclusion, the present study demonstrated that TNC was up-regulated during the inflammatory phase to the early phase of the formation of granulation tissue after MI in humans and modulated the transition from inflammation to tissue healing. In addition to controlling the polarization from M1 to M2 macrophages, the inhibitory effects of TNC on lymphangiogenesis may delay the resolution of inflammation, which affects adverse post-infarct remodeling in the chronic phase.

## 4. Materials and Methods

### 4.1. Materials

All autopsies were performed from 1 January 2007 to 28 February 2017. Autopsy hearts were obtained from 20 patients (male: 13, female: 7) who died from 5 h to 7 years after the onset of MI. Ethics committee approval and informed consent from the patient’s family were obtained, mentioned below in the Institutional Review Board Statement and Informed Consent Statement. The mean age of patients was 81.8 ± 5.9 years (range 65–90 years old), and the mean weight of hearts was 363.1 ± 73.3 g (range 222–487 g). PCI was not performed because the selected cases in the present study consisted of patients who had conditions like severe dementia or advanced malignant tumors (such as glioblastoma or prostate cancer with multiple metastases) or the case when patient or family consent for treatment could not be obtained.

### 4.2. Tissue Preparation

Myocardial tissues were fixed with 10% neutral-buffered formalin and embedded in paraffin. All sections were cut at a thickness of 3–5 μm and placed on glass slides. Thin sections were subjected to hematoxylin-eosin (HE) or elastica-sirius red (ESR) staining. Collagen fibers stained with sirius red were observed by polarization microscopy as previously described [97].

### 4.3. Immunohistochemistry

Immunohistochemical staining was performed with primary antibodies against CD31 (JC70A, M0823, Dako, Hovedstaden, Denmark, dilution 1:150), podoplanin (D2-40, 413451, NICHIREI BIOSCIENCE, Tokyo, Japan, dilution already adjusted), and alpha-smooth muscle actin (αSMA; ab5696, Abcam, Cambridge, UK, dilution 1:300) after antigen retrieval (citrate buffer solution pH 6.0, 121 °C for 1 min). We also used the antibody against TNC (4F10TT, 10337, Immuno-biological Laboratories, Fujioka-Shi, Japan, dilution 1:1000) after antigen retrieval (0.05% pepsin, 37 °C for 10 min) as previously described [98].

### 4.4. Analysis of Vascular Density in Each Phase

The phase in each of the 20 tissues was histologically categorized according to the above definitions. Ten independent fields in each phase area were observed using an optical microscope (BX-51, Olympus, Japan), camera adaptor (U-TV0.5XC-3, Olympus, Tokyo, Japan), and camera monitor (Diamondcrysta RDT196LM, Mitsubishi Electric, Tokyo, Japan). Lymphatic vessels were counted as “one” when they had a lumen composed of endothelial cells that were positive for podoplanin. Lymphatic vessels were sometimes positive not only for podoplanin but also CD31. Therefore, blood vessels were counted as the number of CD31-positive lumens—the number of podoplanin-positive lumens. The numbers of blood and lymphatic vessels were counted in one optic field (0.55 mm^2^) and averaged over 10 fields to obtain the vascular density for each phase. Vascular density (/mm^2^) in each phase was statistically analyzed by a one-way ANOVA using GraphPad Prism 9.0.1 for Mac (GraphPad Software, San Diego, CA, USA).

### 4.5. TNC Treatment of LECs

LECs (CC-2810, Lonza, Basel, Switzerland) were cultured to confluence in EGM2 medium (CC-3162, Lonza, Basel, Switzerland) enriched with 15% fetal bovine serum (FBS). Confluent LECs were incubated in medium free of growth factors and containing 0.5% FBS for 16 h in 6-well plates. TNC was purified from conditioned medium of the U-251MG human glioma cell line [30]. LECs were then exposed to EGM2 medium supplemented with 15% FBS in the presence or absence of TNC (10 μg/mL) for an additional 16 h.

### 4.6. RNA Extraction and Sequencing

Total RNA was extracted from LECs using Trizol (Thermo Scientific, Waltham, MA, USA), following the manufacturer’s guidelines. RNA yields and A260/280 ratios were measured using a NanoDrop ND-100 spectrophotometer (Thermo Scientific, Waltham, MA, USA), while RNA integrity numbers were evaluated with the 2100 Bioanalyzer (Agilent Technologies, Santa Clara, CA, USA). Only RNA samples with integrity numbers > 8 were included in subsequent analyses. Samples were sent to Osaka University’s Research Institute for Microbial Diseases, NGS Core for the generation of cDNA. Sequencing was performed using the Illumina NextSeq High Output kit on the Illumina NovaSeq 6000 system sequencer. Sequenced reads were aligned to the human reference genome (GRCh37) using HISAT2 (version 2.2.1), in conjunction with SAMtools (version 1.7). FeatureCounts (version 2.0.1) was employed to count reads on exons and generate a count matrix. iDEP.96, a web application for an RNA-seq analysis, was used to conduct normalization and statistical tests on differentially expressed genes. An enrichment analysis was performed using ShinyGO 0.772, a web application for an advanced analysis. Thresholds were established at a 1.5-fold change and adjusted *p*-value < 0.05.

## Figures and Tables

**Figure 1 ijms-24-10184-f001:**
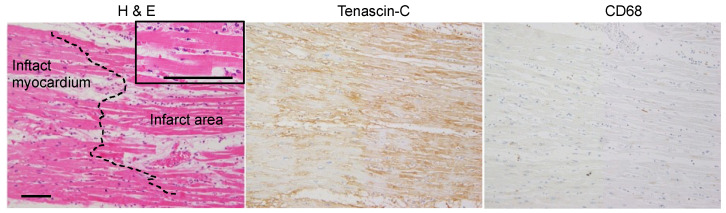
Histology and immunostaining of serial section border zone myocardium at inflammatory phase (case 15). The coagulative necrotic cardiomyocytes with eosinophilic cytoplasm and no nuclei and neutrophil infiltration are seen (in enlarged area). Scale bar = 100 μm.

**Figure 2 ijms-24-10184-f002:**
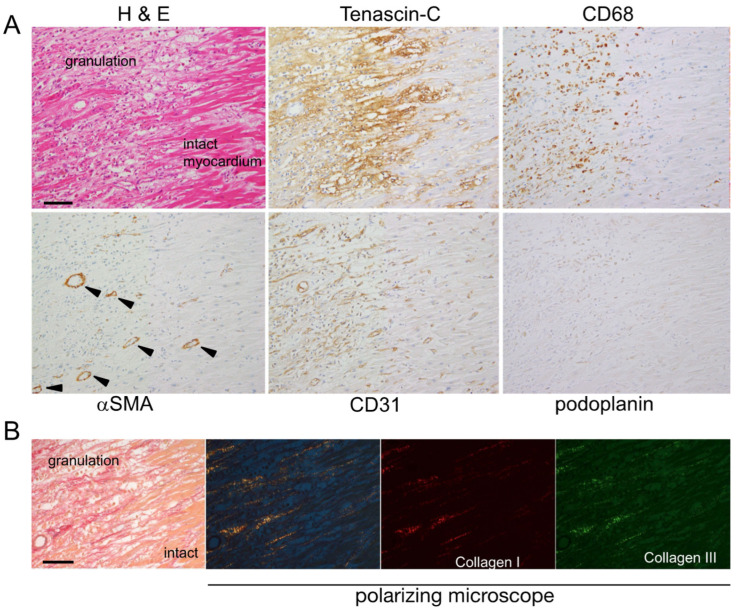
(**A**) Histology and immunostaining of serial sections of myocardial lesion at the early granulation phase, five days after the onset (case 6). CD31-positive angiogenic sprouting and CD68-positive macrophage infiltration are noted in the TNC-positive area at the border zone. Arrowheads indicate αSMA-positive blood vessels, but very few αSMA-positive fibroblasts are seen. Scale bar = 100 μm. (**B**) Collagen fibers observed under the polarizing microscope of the elastica Sirius red-stained myocardial section. With the use of band-pass filters, the red fibers are composed mainly of collagen type I, and the green fibers are composed mainly of collagen type III. A few thin collagen fibers are seen in the granulation tissue. Scale bar = 100 μm.

**Figure 3 ijms-24-10184-f003:**
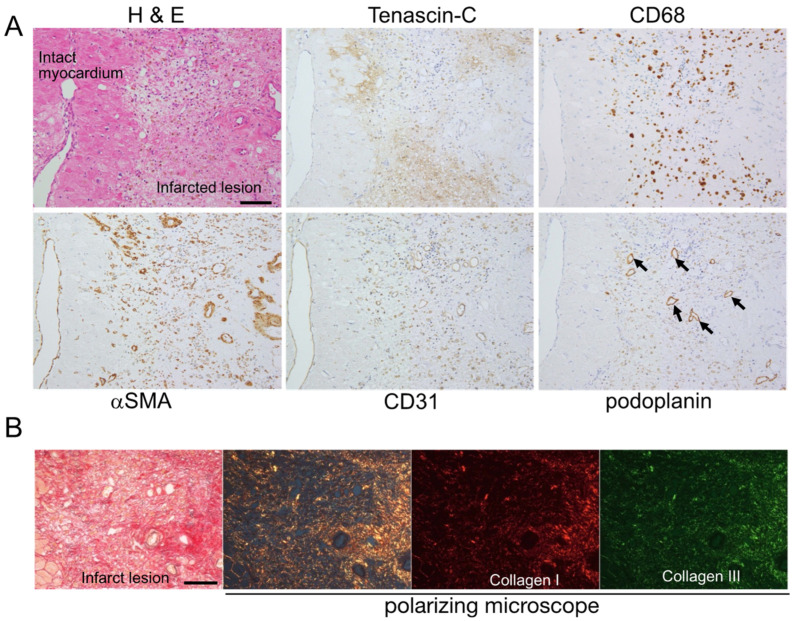
(**A**) Histology and immunostaining of serial sections of myocardial lesions at the fibrogenic phase, ten days after the onset (case 2). Arrows indicate lymphatic vessels in the TNC-negative area. Numerous CD31-positive blood vessels and αSMA-positive myofibroblasts are seen. Scale bar = 100 μm. (**B**) Collagen fibers observed under the polarizing microscope of the elastica Sirius red-stained myocardial section. Randomly arranged thin and short collagen fibers are seen in the infarct area. Scale bar = 100 μm.

**Figure 4 ijms-24-10184-f004:**
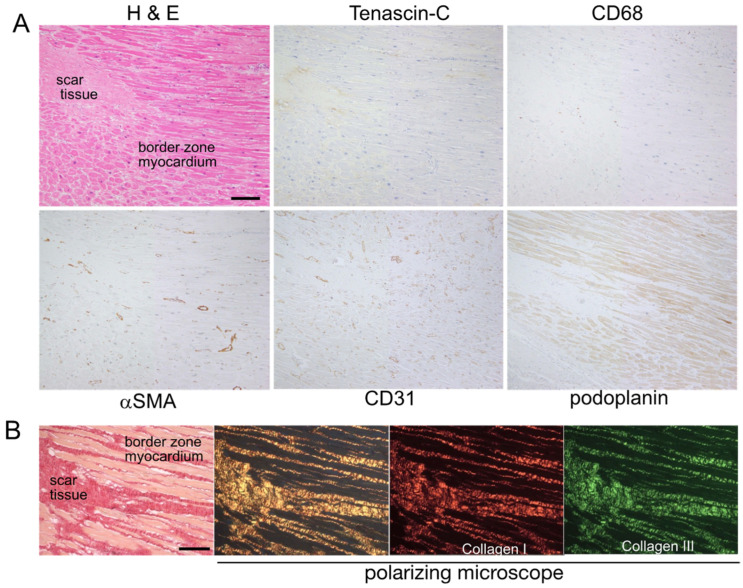
(**A**) Histology and immunostaining of serial sections of myocardial lesion at the scar phase (case 19). No TNC staining is seen at this stage. Macrophages (CD68 positive), myofibroblasts (αSMA positive), blood vessels (CD31 positive), and lymphatic vessels (podoplanin positive) are significantly reduced. Scale bar = 200 μm. (**B**) Collagen fibers observed under the polarizing microscope of elastica sirius red-stained myocardial sections. Well-organized thick collagen fibers are seen in the scar tissue. Scale bar = 100 μm.

**Figure 5 ijms-24-10184-f005:**
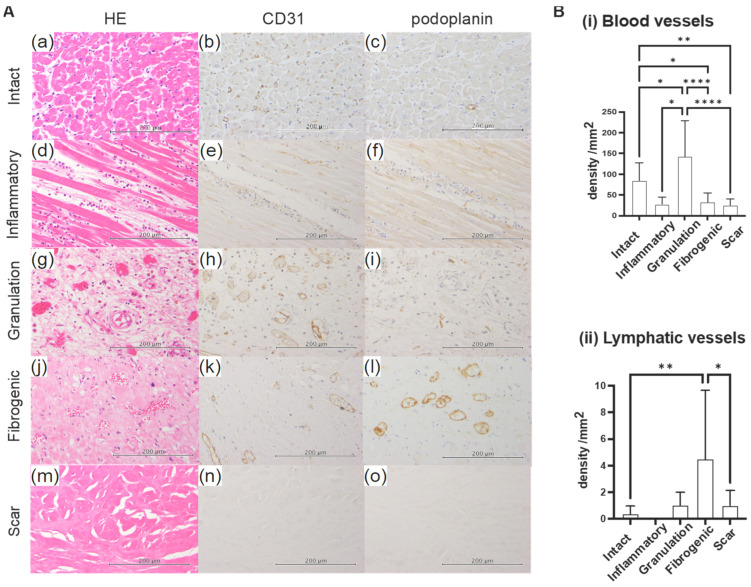
(**A**) Representative photos of blood and lymphatic vessels in each phase. (**a**–**c**) Intact area: Numerous CD31-positive capillary vessels are observed around cardiomyocytes, while fewer podoplanin-positive lymphatic vessels are present. (**d**–**f**) Inflammatory phase: Cardiomyocytes show coagulative necrosis. Neutrophils infiltrate an edematous interstitium. Blood vessels and lymphatic vessels disappear. (**g**–**i**) Granulation phase: New CD31-positive capillary vessels proliferate. A few podoplanin-positive cells are observed. (**j**–**l**) Fibrogenic phase: The number of lymphatic vessels increases, while that of blood vessels decreases. (**m**–**o**) Scar phase: The only remaining tissue is a fibrotic scar, which does not contain any cellular components. Blood vessels and lymphatic vessels are no longer present. Scale bar = 200 μm. (**B**) Densities of blood vessels and lymphatic vessels in each phase. (**i**) Blood vessels: Blood vessels decrease in the inflammatory phase and peak in number in the granulation phase. They then gradually decrease during the fibrogenic and scar phases. (**ii**) Lymphatic vessels: Lymphatic vessels are rarely present in intact areas. They disappear in the inflammatory phase, gradually increase in the granulation phase, and peak in number in the fibrogenic phase. They then disappear in the scar phase. Data are expressed as the mean ± SD. The significance of differences was assessed with a one-way ANOVA test. The numbers in each phase are as follows: Intact *n* = 19, inflammatory *n* = 3, granulation *n* = 5, fibrogenic *n* = 14, scar *n* = 12. * *p* ≤ 0.05, ** *p* ≤ 0.01, **** *p* ≤ 0.0001.

**Figure 6 ijms-24-10184-f006:**
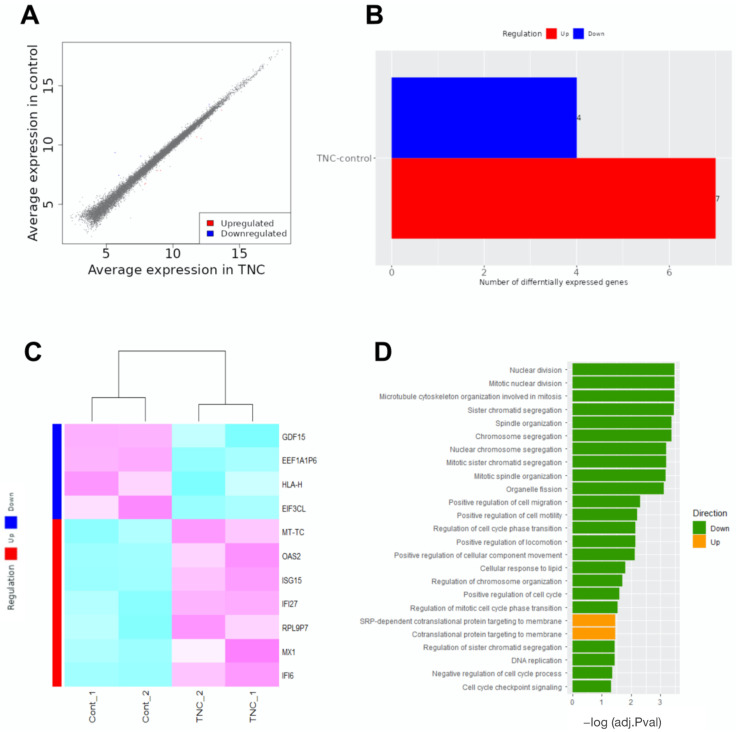
RNA-seq analysis of TNC-modulated genes. (**A**) Scatter plot displaying differentially expressed genes (DEGs) from the RNA-seq analysis of control and TNC-treated LECs. Up- and down-regulated genes are represented in red and blue, respectively. Values are shown as the log2 of normalized counts. (**B**) RNA-seq comparison identifying seven positively regulated genes (red) and four negatively regulated genes (blue). (**C**) Heatmap illustrating the expression profiles of positively and negatively regulated genes following the TNC treatment. The red bar represents Up-regulated, and the blue bar represents Down-regulated gene groups. The darker magenta color indicates higher gene expression and the darker cyan color indicates lower expression. (**D**) DEGs categorized based on their gene ontology (GO) annotation in biological processes, using the generally applicable gene-set enrichment (GAGE) method.

**Table 1 ijms-24-10184-t001:** Characteristics of autopsied patients after MI.

Patient No.	Age	Gender	Heart Weight (g)	Time from MI Onset to Autopsy	STEMI or NSTEMI	EF (%)	Direct Cause of Death
1	88	M	376	5 h	NSTEMI	N/A	MI
2	85	F	308	10 days	NSTEMI	N/A	MI
3	80	F	307	N/A	N/A	N/A	pneumonia
4	80	M	330	N/A	N/A	N/A	MI
5	65	M	395	2 days	STEMI	N/A	MI
6	83	M	487	5 days	STEMI	N/A	MI
7	82	M	278	30 days	STEMI	50	pneumonia
8	88	F	362	14 days	NSTEMI	33	MI
9	86	M	305	16 days	N/A	15	MI
10	72	M	363	45 days	N/A	N/A	sepsis
11	79	M	222	N/A	N/A	N/A	glioblastoma
12	81	M	344	7 years	N/A	N/A	gastrointestinal bleeding
13	80	M	406	N/A	N/A	N/A	prostate cancer
14	84	F	403	2 years	N/A	N/A	SAH
15	79	M	472	N/A	N/A	N/A	aspiration pneumonia
16	83	F	335	16 days	NSTEMI	N/A	NOMI
17	90	M	445	18 days	NSTEMI	N/A	MI
18	87	F	479	2 days	N/A	15	MI
19	87	M	259	20 days	N/A	N/A	organizing pneumonia
20	76	F	386	N/A	N/A	N/A	lethal arrhythmia

Abbreviation: EF = ejection fraction, MI = myocardial infarction, N/A = not available, NOMI = non-occlusive mesenteric ischemia, NSTEMI = non-ST-elevation myocardial infarction, SAH = subarachnoid hemorrhage, STEMI = ST-elevation myocardial infarction.

## Data Availability

The data that support the findings of this study are available from the corresponding author (K.I.-Y.), upon reasonable request.

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
