# Peer review of "Tenascin-C in Tissue Repair after Myocardial Infarction in Humans"

_ijms, 2023, doi:10.3390/ijms241210184_

Round 1

Reviewer 1 Report

Having saying that said, I can say, after a careful reading of the submitted article, that it is an excellent scientific article, well written, with an original content of unquestionable interest, and that it meets the requirements demanded by the International Journal of Molecular Science – MDPI. Despite my opinion transcription regarding what is requested in the electronic form from International Journal of Molecular Sciences MDPI, I would like to add the following comments: The data described point to a window of therapeutic opportunities that this pathology, so frequent and so important, demands of all of us. Thus, in Abstract, there is an important reference to several aspects that make this article on acute myocardial infarction in humans very interesting: - Line 23: “focus on lymphangiogenesis, the role of which has recently been attracting increasing attention as a mechanism to resolve inflammation”; - Line 24: “The direct effects of TNC on human lymphatic endothelial cells were also assessed by RNA sequencing. - Lines 25-27: “The results obtained support the potential roles of TNC in the regulation of macrophages, sprouting angiogenesis, the recruitment of myofibroblasts, and the early formation of collagen fibrils during the inflammatory phase to the early granulation phase of human MI. - Lines 32-33: “The present results indicate that TNC induces prolonged over-inflammation by suppressing lymphangiogenesis, which may be one of the mechanisms underlying adverse post-infarct remodeling”   Thus, it is an original article with well-structured chapters with scientifically correct content and appropriate citations. There are, however, some aspects that deserve changes that are referenced in red: (in red purposals to be potentially changed and also to be praised) as well as others that should be highlighted because they are very important in this type of scientific articles (and therefore also to be praised).   Title/authors: Ok Abstract - Ok 1. Introduction: Excellent and logical sequence in view of the pathophysiology and expressed in the introduction...because (emphasis on important approach taken): Lines 38-40: “Acute myocardial infarction (MI) results from severe ischemia due to coronary arterial occlusion”/obstruction, (must also include obstruction such as that which occurs in acute myocardial infarctions without ST elevation) - Lines 101-105: “Therefore, we herein examined human autopsy samples at different stages after MI with immunostaining and analyzed the localization of TNC during human myocardial repair with a focus on lymphangiogenesis. Furthermore, the direct effects of TNC on lymphatic endothelial cells (LECs) were investigated using RNA sequencing”. Lines 106-108: “The results obtained suggest the involvement of TNC in the transition from inflammation to the early stage of formation of granulation tissue and its potential to suppress lymphangiogenesis” consistent with the conclusions obtained in this original investigation Introduction: Excellent review of the potential involvement of TNC in post-AMI noxious remodeling…innate immunity….lymphangiogenesis… well-founded in the literature that references in the bibliography that is appropriate to the subject. - Lines 102-104: good perspective on the approach: “Therefore, we herein examined with immunostaining and analyzed the localization of TNC during human myocardial repair with a focus on lymphangiogenesis. Furthermore, the direct effects of TNC on lymphatic endothelial cells (LECs) were investigated using RNA sequencing”.   - 2. Results - 2.1. Expression mapping of TNC during tissue repair after MI in humans Lines 110-114: “4 phases based on the appearance of inflammatory cells, angiogenesis, and collagen fibers: the inflammatory phase, granulation phase, fibrogenic phase, and scar phase.” Very important approach.   - In this chapter of results: a table is missing showing the main characteristics of the population studied (age, gender; STEMI, NSTEMI numbers; maximum troponin value; day of autopsy after the onset of symptoms…left ventricle ejection fraction and Tenascin-C expression…)

 The figure (1)does not allow us to understand what the authors intend to demonstrate; does not correlate with the text present on lines 126 to 131: caption is inadequate/incomplete; - Thus, Figure 1 lacks: (and the others) a better explanatory caption (asking for a more reasoned opinion from Pathological Anatomy experts) (such as: pointing out the nuclei of cardiomyocytes explanation why cell nuclei are potentially not seen); slides obtained how many days after AMI….?   - Line 139: “Early granulation phase:” clearly improve subtitle…Imunnostainning with …antibodies…   - Figure 5 …excellent and well commented …allows us to understand the dynamics of the expression of blood vessels and lymphatics in the different post-acute infarction phases in Humans Line .258-296: quantification of blood vessels and lymphatics (is it possible to show the post-infarction days to which each phase corresponds (inflammation, granulation…scar…?)   Line 298: 2.3. RNA sequencing – very specific…must be reviewed by another Referee…   3. Discussion Line 346- 349: ” Myocardial tissue repair after MI progresses in a time-dependent manner and is often divided into 3 phases: the inflammatory phase, proliferative phase, and maturation phase. Generally in humans, the inflammatory phase lasts up to 4 days and the proliferative phase occurs from days 5 to 14 after MI. [12], “we categorized histopathological findings into 4 phases based on the appearance of inflammatory cells, angiogenesis, and collagen fibers: the inflammatory phase, granulation phase, fibrogenic phase, and scar phase”. potential contradiction with what was previously said     - Line 358: “..who did not receive reperfusion therapy. The mapping of TNC expression …”; what were the reasons for patients not receiving reperfusion therapy; what were the years of the autopsies?   - Line 478-480 4. Materials and Methods 4.1. Materials “ Autopsy hearts were obtained from 20 patients (male:13, female:7) who died from 1 day to 2 or 3 months after the onset of MI.” …at 2 -3 months the expression of Tenascin and the histopathological changes are the same?? -          Lines 481-482: - The mean age of patients was 81.8±5.9 years 481 (range 65-90 years old)…”, in the hearts of the younger population during an AMI, lymphangiogenesis and Tenascin expression…is it the same? obtained in this study (topic for discussion)   - Line 584: - 5. Conclusions: ok depending on the data found - Conclusions: “Therefore, we herein examined human autopsy samples at different stages after MI with immunostaining and analyzed the localization of TNC during human myocardial repair with a focus on lymphangiogenesis. Furthermore, the direct effects of TNC on lymphatic endothelial cells (LECs) were investigated using RNA sequencing. The results obtained suggest the involvement of TNC in the transition from inflammation to the early stage of formation of granulation tissue and its potential to suppress lymphangiogenesis”, which fits into the set of original and important results for publication   Line 584: References: 1. Reed, G.W.; Rossi, J.E.; Cannon, CP. Acute myocardial infarction. Lancet 2017, 389, 197-210. 585 2. Zhang, R.Y.K.; Cochran, B.J.; Thomas, S.R.; Rye, K.A. Impact of Reperfusion on Temporal Immune Cell Dynamics 586 After Myocardial Infarction. J Am Heart Assoc 2023, 12, e027600. The References are correct, according to the requirements and appropriate to the topic in publication. However, the year of publication must appear in bold.

Reviewer 2 Report

The topic of the study is interesting and the Authors extensively discuss what is already known in the literature and the results obtained.

However, it appears necessary and indispensable to make some substantial improvements.

First of all, improve the Materials and Methods section which appears not very detailed; here are some suggestions:

-          Define the study: was the study retrospective or prospective? Observational or interventional? Was the autopsy performed for the purpose of the study? And if so, has ethics committee approval been obtained? Was informed consent obtained from family members?

-          Better characterize the study population: why did the patients not receive reperfusion therapy which is the first choice in case of myocardial infarction? Was it STEMI or NSTEMI? Was myocardial infarction the cause of death? Were there other causes of active inflammation in these patients?

-          Finally: the Materials and Methods section must be the n.2 of the paper, i.e. before the results.

Second: a major revision of the text is necessary because some parts seem copied and pasted from the instructions for the authors of the journal, other times they are repetitions; I mention a few:

-          lines 473-477

-          lines 539- 553

-          lines 468-473 are exactly the same as lines 555-560, i.e. the "Conclusions" paragraph

-          lines 106-108 should be deleted as the introduction section should contain no results

Minor editing of English language required.

Round 2

Reviewer 2 Report

Looking at table 1, it seems that myocardial infarction is a current condition only in a few cases, while in others it is an anamnestic fact.

Thus the troponin values are not helpful, if not confounding: I suggest deleting that column.

It is appropriate to make a minor revision of the English language.
